# Crosstalk between COVID-19 Infection and Kidney Diseases: A Review on the Metabolomic Approaches

**DOI:** 10.3390/vaccines11020489

**Published:** 2023-02-20

**Authors:** Reshma Murali, Uddesh Ramesh Wanjari, Anirban Goutam Mukherjee, Abilash Valsala Gopalakrishnan, Sandra Kannampuzha, Arunraj Namachivayam, Harishkumar Madhyastha, Kaviyarasi Renu, Raja Ganesan

**Affiliations:** 1Department of Biomedical Sciences, School of Biosciences and Technology, Vellore Institute of Technology (VIT), Vellore 632014, Tamil Nadu, India; 2Department of Cardiovascular Physiology, Faculty of Medicine, University of Miyazaki, Miyazaki 889-1692, Japan; 3Center of Molecular Medicine and Diagnostics (COMMAND), Department of Biochemistry, Saveetha Dental College and Hospitals, Saveetha Institute of Medical and Technical Sciences, Saveetha University, Chennai 600077, Tamil Nadu, India; 4Institute for Liver and Digestive Diseases, College of Medicine, Hallym University, Chuncheon 24252, Republic of Korea

**Keywords:** COVID-19, kidney diseases, metabolomics, pathophysiology

## Abstract

The novel severe acute respiratory syndrome coronavirus 2 (SARS-CoV-2) causes COVID-19, a respiratory disorder. Various organ injuries have been reported in response to this virus, including kidney injury and, in particular, kidney tubular injury. It has been discovered that infection with the virus does not only cause new kidney disease but also increases treatment difficulty and mortality rates in people with kidney diseases. In individuals hospitalized with COVID-19, urinary metabolites from several metabolic pathways are used to distinguish between patients with acute kidney injury (AKI) and those without. This review summarizes the pathogenesis, pathophysiology, treatment strategies, and role of metabolomics in relation to AKI in COVID-19 patients. Metabolomics is likely to play a greater role in predicting outcomes for patients with kidney disease and COVID-19 with varying levels of severity in the near future as data on metabolic profiles expand rapidly. Here, we also discuss the correlation between COVID-19 and kidney diseases and the available metabolomics approaches.

## 1. Introduction

The new type of β-coronavirus, which is an enveloped, single-stranded, positive sense RNA virus, causes pneumonia [1]. In humans, SARS-CoV, SARS-CoV-2, and Middle East respiratory syndrome coronavirus (MERS-CoV) cause severe respiratory syndrome [2,3]. Hong Kong reported that the SARS pandemic affected pediatric and adult kidney transplant recipients, with pediatric patients suffering from less severe symptoms [4]. In 2003, one liver transplant patient with SARS-CoV infection died [5]. One report of two kidney transplant patients found that one died from a progressive respiratory disease and an acute renal injury. In contrast, the other one survived the MERS-CoV infection [6].

A substantial percentage of COVID-19 patients also show signs of kidney disease [7]. SARS-CoV-2 enters cells through the cytoprotease Transmembrane protease serine 2 (TMPRSS2), a factor required for cell entry, and through the angiotensin-converting enzyme II (ACE2) receptors, found throughout the body is highly expressed in the proximal kidney tubules [8,9,10]. It is hypothesized that SARS-CoV-2 may cause kidney injury by entering cells via targeting ACE2, which is widely expressed in the kidneys [11]. SARS-CoV-2 infections are more likely to cause kidney damage than SARS-CoV infections is possibly due to SARS-CoV-2’s increased affinity for the receptor protein ACE2 (approximately 10 to 20 times than that of SARS-CoV) [12,13].

At first, renal involvement was considered insignificant, and the incidence of AKI received little attention [14]. The incidence of AKI was found to be 15% in COVID-19 patients, demonstrating that AKI is prevalent and that the virus can specifically harm the kidneys. In critically ill COVID-19 patients with pre-existing diseases, the incidence of AKI may increase to 25% [15,16]. Patients with COVID-19 who have chronic kidney disease (CKD) and other comorbidities are said to be more likely to develop a severe form of the illness. Thus, they should take special precautions to prevent coming into contact with SARS-CoV-2 [17]. A more significant number of comorbidities were also shown to be associated with the virus’s affinity for the kidney, as COVID-19 induced renal damage directly through virus tropism and indirectly through cytokine storm and resulted in higher mortality associated with kidney injury [18].

Even patients whom COVID-19 hasn’t infected are likely to experience long-term consequences since delayed care for CKD, diabetes, and hypertension is expected to result from the pandemic [19]. However, it is unclear whether COVID-19’s effects are unique or similar to those of other infections or sepsis. In initial reports, COVID-19 is associated with many of the same sequelae as sepsis and acute respiratory distress syndrome (ARDS) [20]. Novel urine and kidney-specific plasma biomarkers may be able to identify the primary underlying causes and identify those individuals who are more at risk for developing CKD following hospitalization for COVID-19 [21]. Pretorius et al. reported that in COVID-19 patients who have recovered from acute COVID-19 symptoms, a novel phenotype known as Long COVID/Post-Acute Sequelae of COVID-19 (PASC) has been identified. Proteomics research revealed large increases in acute phase proteins such as SAA4 and α2AP, indicating that the plasma samples from COVID-19 and PASC patients are highly resistant to breakdown in the presence of trypsin [22]. According to Al-Aly et al. study that evaluated lengthy COVID-19 using electronic health data from the Veterans Health Administration, COVID-19 increased the risk of CKD and that risk was highest in patients with severe disease [23]. Su et al. carried out a comprehensive multi-omic longitudinal study of 309 COVID-19 patients from the time of their first diagnosis to convalescence (2–3 months later), integrating clinical information and patient-reported symptoms and four PASC-anticipating risk factors were resolved upon diagnosis of COVID-19: type 2 diabetes, SARS-CoV-2 RNAemia, Epstein-Barr virus viremia, as well as specific autoantibodies. SARS-CoV-2 and cytomegalovirus specific CD8^+^ T lymphocytes in individuals with gastrointestinal PASC displayed unique dynamics during COVID-19 recovery. Symptom-associated immunological signatures reveal four endotypes with divergent acute severity and PASC [24].

The development of metabolomics, or high-throughput measurement and analysis of metabolites, has made it possible to examine kidney diseases in great detail and laid the groundwork for creating new molecular diagnostic tools for use in nephrology. Small-molecule metabolites observation in the blood or urine are used to diagnose or evaluate kidney damage. The ability to probe the molecular pathways behind kidney disease is made possible by the power of metabolomics [25]. The COVID-19 pandemic has given the use of metabolomics in the diagnosis of infectious diseases a boost. Using metabolomics methods based on measuring volatile organic compounds exhaled by COVID-19 patients, large-scale population screening can be done at point-of-care [26].

The aim of this narrative review is to discuss the possible applications of analytical strategies such as metabolomics in the study of COVID-19, kidney and COVID-19 induced kidney injury and to address the therapeutic options to treat the COVID-19 induced kidney injury.

## 2. COVID-19 and AKI

According to research, people with COVID-19 have an uneven incidence of AKI, ranging from 0.1% to 29% [11,27,28,29]. Early findings from China and Italy showed that the risk of AKI ranged widely from 0.5% to 29%, with most estimates on the lower end. In contrast, USA data has only been limited to critically ill patients in the intensive care units (ICU) in a Seattle hospital, where a 19% incidence of AKI was observed [27,28,29,30,31]. In critically ill COVID-19 patients, AKI, which affects 20–40% of patients sent to ICU, is widespread, according to experience in Europe and the USA [32]. It is viewed as an indication of the severity of the disease and a poor prognostic sign for survival [11,30]. Because earlier serum creatinine levels may not be available and measurement of creatinine values at the time of admission may not accurately reflect renal function before the admission, it is also likely that COVID-19 may underestimate the actual overall burden of AKI [33,34,35]. AKI among hospitalized COVID-19 patients in the USA may vary from 28% to 46%, with a significant risk of in-hospital mortality. Compared to patients with normal kidney function, those who develop AKI have a worse prognosis [36,37].

### 2.1. Pathogenesis of AKI in COVID-19 Patients

COVID-19 typically causes respiratory failure and hypoxemia, but kidney involvement is also observed. AKI may result from SARS-CoV-2 infection via several pathophysiological routes. Though kidney function is drastically decreased, acute tubular damage is common and usually minor injuries to the tubules are probably exacerbated by systemic hemodynamic instability. Endothelial injury and microvascular thrombi have also been linked to kidney injury, as have tissue inflammation and local immune cell infiltration. In addition, people with severe COVID-19 have been reported to have a defective type I interferon (IF) response. Given these findings, it would be helpful to gain insight into the potential pathophysiological pathways of COVID-19-associated AKI to inform therapeutic methods [34,38,39,40,41,42,43].

Inflammatory and immunological responses are all potential contributors to the development of COVID-19 AKI [44,45]. Renal tropism of the virus, which could occur through direct infection, has also been hypothesized but is still debatable [46]. Kidney injury and functional impairment may also be caused by frequent non-specific variables [38].

#### 2.1.1. Overactivation of Angiotensin II Pathway

Kidney injury molecule 1 (KIM1) was discovered to be a receptor for SARS-CoV-2 in tubule epithelial cells [47], even though ACE2 is generally accepted as the traditional receptor by which SARS-CoV-2 obtains entry into cells. TMPRSS2 is expressed by kidney cells; the proteolytic cleaving enzyme of ACE2 is required for viral entry [18,48]. While ACE2 is primarily expressed in the proximal tubules, TMPRSS2 colocalizes to many kidney regions. It is most highly expressed in the distal tubules [49,50,51,52,53]. SARS-CoV-2 binding to human ACE2 is thought to result in the downregulation of ACE2 [54], resulting in elevated angiotensin II (Ang-II) and reduced Ang (1–7) [55,56,57,58]. Both interleukin -6 (IL-6) and IL-8 levels were significantly decreased by recombinant human soluble ACE2 [58,59].

Even while ACE2 plays a critical role in the kidney in converting Ang-IIinto Ang (1–7), it appears that Ang (1–7) production in the plasma and lungs occurs mostly independently of ACE2 [60]. Notably, soluble ACE2 levels in the blood are pretty low [61,62]. Though ACE2 polymorphisms have been reported, their association with COVID-19 AKI remains unknown [63]. Future research should investigate whether these genetic variants are linked to distinct patterns of harm [38]. To what extent low levels of Ang-IIare related to poor outcomes in critically sick patients [64,65] is unclear and may depend on the severity of the disease and whether or not it constitutes an adaptive response to shock. In a small study of critically ill COVID-19 patients, researchers found a link between AKI and elevated plasma renin levels, associated with decreased Ang-II activity [65,66]. Individuals with COVID-19 and ARDS have lower Ang-II levels than patients with milder diseases, suggesting the possibility of the exact mechanism [67].

#### 2.1.2. Dysregulated Immune Responses in COVID-19

The most severe variants of COVID-19 are known to have CD4^+^ and CD8^+^ T-lymphopenia as distinguishing characteristics. Some studies suggest that insufficient adaptive immunity can also contribute to poor outcomes in this disease. It has also been noted that eosinophils, natural killer (NK) cells, and plasmacytoid dendritic cells (a primary source of IF-α) have been depleted. The transcription factor Nrf2 controls cellular antioxidant responses [68]. While it is still premature to draw firm conclusions about the protective function of Nrf2 activation in COVID-19-related AKI, findings from experimental AKI in other situations lend credence to this notion. Increased expression of Nrf2 in T-cells protects against renal functional and histological damage after ischemia-reperfusion injury in a mouse model. This benefit is coupled with decreased tumor necrosis factor-α (TNF-α), IF-γ, and IL-17. On the contrary, a lack of Nrf2 makes tissues more vulnerable to ischemic and nephrotoxic damage, suggesting a potential therapeutic role for this transcription factor [68].

COVID-19-infected patients have been shown, like those infected with other viruses like Ebola [69], to display a spectrum of phenotypic responses in terms of their humoral immune system, with some showing a reduction in the number of circulating memory B cells. In contrast, others show an increase in the number of circulating plasmablasts. Immunosenescence may be at the root of an autoimmune reaction against a soluble version of ACE2. Histological examination of lung tissue from the deceased corroborated the activation of complement components in endothelial cells by purified anti-ACE2 IgM, demonstrating the angiocentric pathophysiology of advanced disease [70]. As ACE2 is a negative regulator of ACE, it is possible that the development of anti-ACE2 autoantibodies has a role in COVID-19 pathogenesis, resulting in tissue edema, inflammation, and damage exacerbated. ACE2 is expressed widely throughout the body, including the Kidney. This notion, however, is still in its early stages and has yet to be tested. Further, it is possible for ACE2 and ACE receptors [71] to cross-react due to their same homology [72,73,74].

#### 2.1.3. Rhabdomyolysis

Rhabdomyolysis is a muscle breakdown disorder that can cause renal impairment. An underlying ailment commonly causes it. A viral or bacterial illness often triggers rhabdomyolysis. SARS-CoV-2 has recently posed a global health crisis. Preceding rhabdomyolysis, there haven’t been many documented examples of SARS-CoV-2 infection [75].

Patients in the hospital have been found to have elevated levels of albuminuria and proteinuria, as well as renal inflammation and edema. It has been observed that the kidney is a common target for the virus. Although the precise mechanisms of renal involvement remain unknown, a pathway has been outlined that integrates several potential contributors [76]. Acute renal tubule injury due to hemodynamic instability is probably the critical mechanism for AKI in patients with severe COVID-19. Direct renal dysfunction, such as collapsing glomerulopathy, has been recorded [77,78,79].

Research published in the last several years suggests that COVID-19 may be linked to rhabdomyolysis, either as a late-stage consequence or presenting issue [80,81,82]. Recent research indicates that the viral invasion of myocytes during influenza infection is the direct cause of rhabdomyolysis [77]. Whether rhabdomyolysis caused by COVID-19 results from indirect viral invasion or direct muscle injury triggered by inflammatory mediators such as cytokines is unknown but likely [77], it is not possible to rule out the possibility that skeletal muscle damage is not caused by a viral infection, as viral elements are not the only contributors to rhabdomyolysis. Hydroxychloroquine and oseltamivir have been linked to rhabdomyolysis, as described in pre-COVID-19 literature [83]. Using these medications to treat COVID-19 raises the possibility that they could cause rhabdomyolysis [83]. Only a few isolated cases of rhabdomyolysis have been reported previously [84].

There was rhabdomyolysis in a severe COVID-19 patient accompanied by fatigue and pain in the lower limbs [85]. Damaged cells release myoglobin, causing AKI, a serious complication of severe rhabdomyolysis [86]. AKI caused by COVID-19 infection results in hemosiderin deposition and pigment casts in tubules, which confirms rhabdomyolysis [87]. Kidney failure and rhabdomyolysis caused by COVID-19 currently have no cure. All changes in status, including those shown in blood and urine indicators, must be considered while tailoring treatment to individual symptoms. Restoring the equilibrium of the renin-angiotensin-aldosterone system (RAAS) may be an effective treatment strategy for reversing the renal dysfunction caused by viral activity. It is essential to tailor therapy to each patient and ensure the benefits outweigh any risks [88,89].

#### 2.1.4. Sepsis

Sepsis-associated AKI shares similarities with COVID-19 AKI, which is of interest. While renal blood flow can be lower or higher than normal rates, sepsis-associated AKI is defined by a decline in GFR [90]. As a result, several professionals suggest that viral sepsis is important in the pathophysiology of COVID-19. Regional inflammation, microvascular changes, and haemodynamic changes (including glomerular shunting, activation of tubuloglomerular feedback, and increased interstitial and thus intratubular pressure) are all factors that lead to sepsis-associated AKI [91,92]. A typical consequence of seriously ill patients is septic AKI, which is brought on by alterations in renal hemodynamics, immune cell activation, a large-scale release of inflammatory chemicals, and endocrine instability [93,94].

### 2.2. Pathophysiology of COVID-19 and AKI

#### 2.2.1. Tubular Injury

COVID-19 causes ARDS, characterized by localized inflammation and the recruitment of immune cells like macrophages, effector T-cells, and polymorphonuclear neutrophils. The immune cells release IFs for viral clearance. The lungs will release cytokines in response to pathogen-associated molecular patterns (PAMPs) and damage-associated molecular patterns (DAMPs), increasing inflammation and tissue damage. Neutrophil extracellular traps (NETs), produced by active neutrophils, may impact the local inflammatory response, pathogen clearance, and thrombosis. Increased tissue edema causes an increase in renal interstitial pressure, which damages the tubules [38]. The majority of COVID-19 patients with AKI had acute tubular injury characterized by primarily mild focal acute tubular necrosis, according to the most recent data available [45,87,95,96]. In addition to COVID-19 infection, other factors can also result in tubular injury, including local inflammatory processes, which can release cytokines and activate the complement system, medication-induced tubular injury, rhabdomyolysis, hypovolemia brought on by fluid losses from fever or diarrhea, hypotension or septic shock, pro-coagulant status, and activation of the renin-angiotensin-aldosterone system [97] (Figure 1).

#### 2.2.2. Endothelial Activation and Microvascular Injury

The endothelium is the interface between the blood and the body tissues. Hence, endothelial dysfunction and systemic inflammatory response affect most of the organs and systems in the body [98]. Common symptoms of a pro-coagulant state and disseminated intravascular coagulation include an increased lactate dehydrogenase level, extended prothrombin and partial thromboplastin times, thrombocytopenia, and the potential for deep vein thrombosis or pulmonary embolism [99]. There are direct and indirect mechanisms by which COVID-19 causes AKI, including endotheliitis, thrombosis, and abnormal glucolipids [100]. COVID-19 is associated with microvascular and macrovascular complications, including myocardial infarction and stroke, due to endothelial dysfunction [101]. Numerous endothelial dysfunctions, such as reduced nitric oxide (NO) bioavailability, oxidative stress, endothelial injury, hyperpermeability, disruption of the glycocalyx and barrier, inflammation/leukocyte adhesion, hypercoagulability, senescence, endothelial-to-mesenchymal transition (EndoMT), and thrombosis, among others, have been linked to SARS-CoV-2 infection. COVID-19 is therefore categorized as an endothelial and microvascular disease [100].

#### 2.2.3. Podocyte Injury

Podocytes are the terminally differentiated cells of the renal glomerulus, and they play a crucial role in maintaining the integrity of the kidney filter [102]. Among COVID-19 patients, collapsing glomerulopathy is the most common glomerular disease associated with Apolipoprotein L1 (APOL1) gene polymorphisms, especially in African Americans [15]. Viral infections increase APOL1 expression, which activates IFs and toll-like receptors, dysregulating podocytes and glomeruli [103,104]. While COVID-19 patients have reported sporadic instances of minimal change disease and focal segmental glomerulosclerosis, the collapsing glomerulopathy has been linked to various viral infections associated with IF release, including HIV, cytomegalovirus, Epstein-Barr virus (EBV), and Parvovirus B19 [105,106].

### 2.3. Inflammatory Responses

Following the SARS-CoV-2 infection, several changes in innate and adaptive immune responses have been observed. Soon after the natural immune response is triggered, the adaptive immune system begins to manage viral infection through antibody-producing B-cells, CD4^+^ T cells that aid in viral clearance, and CD8^+^ T cells that defend against the virus through a variety of cytokines. TNF-α and FAS, are inflammatory mediators that can directly harm renal endothelial and tubule epithelial cells by binding to their particular receptors [107,108]. The production of PAMP and DAMP molecules may also activate complement and inflammatory pathways, which may cause the release of extrinsic coagulation-related tissue factors and pro-coagulant substances [34]. The prototypical DAMP, HMGB1, activates the inflammasome in COVID-19 patients’ inflammatory response [37].

### 2.4. Cytokine Storm Syndrome

Organ failure, increased immune system cell proliferation, T cells, macrophages, and NKcells, and increased production and release of inflammatory cytokines all occur in cytokine storm syndrome (CSS) [109]. A severe SARS CoV-2 infection is characterized by CSS, which is characterized by elevated blood levels of cytokines, including IL-1β, IL-6, IL-2R, and TNF-α, released by various cell types, including endothelial cells [28]. IL-6 is a key cytokine involved in multiple malfunctioning organs, including AKI. IL-6 levels are elevated in COVID-19 patients and are substantially associated with adverse clinical outcomes, such as ICU hospitalization, ARDS, and death [42,59,110]. The severity of the inflammatory storm that causes ARDS is also influenced by genetic predisposition linked to genes for ACE2, TNF-α, VEGF, IL-10, etc. [111]. AKI in ARDS can be caused by five factors: hemodynamic instability, hypoxemia/hypercapnia, inflammation, acid-base dysregulation, and neurohormonal effects [112].

## 3. COVID-19 and CKD

There is still some uncertainty over whether or not CKD is a cause or a consequence of severe COVID-19. Five primary studies and one systematic review revealed that patients with CKD had a higher frequency of severe COVID-19 disease [113,114,115,116,117,118]. Inducing a prothrombotic condition, COVID-19 can raise the danger of thromboembolic events in the veins and arteries [119,120]. There was no CKD definition provided in that study. Among critically sick COVID-19 patients, the occurrence of thromboembolic events was identical regardless of CKD status [121,122].

Unfortunately, COVID-19 severe infection was not clearly defined and presumably varied across investigations. The pooled data from a single meta-analysis indicated a significant connection between CKD and the severity of COVID-19 [17]. It is essential to highlight that there was inconsistency in the results that informed some conclusions. The effect estimates for the requirement for ICU. For instance, admission and poor outcomes in patients with CKD and COVID-19 varied significantly among the selected primary studies. In addition, there were discrepancies in the number of primary researches included in the various reviews published.

Furthermore, it is possible that discrepancies in results can be attributed to the fact that certain studies did not account for all relevant confounders. Results from this and other research highlight the need for COVID-19 prevention and management guidelines tailored to the needs of CKD patients. Although patients on maintenance hemodialysis who have been vaccinated have mounted an immune response, their antibody titers are lower than those of healthy controls, according to recent research [123,124]. Research into the preventive effects of the COVID-19 vaccine in people with CKD is currently being conducted. Patients with CKD and COVID-19 were shown to have a significantly higher risk of mortality and hospitalization, as demonstrated by numerous studies. Infection and other adverse effects may be more common in people with CKD. However, it is unclear to what extent this is the case, and study results are inconsistent. The findings highlight the importance of prioritizing COVID-19 immunization and intensive treatment for patients with CKD. Understanding the mechanism underlying the effect of CKD on COVID-19 results might improve the care of these individuals [113,115].

## 4. Metabolomics, COVID-19, and Kidney Injury

Compared to the proteome or transcriptome, metabolomics is more accurate in measuring a cell’s metabolic state [125]. In contrast to current PCR and antibody tests, metabolomics studies will help measure and assess the effects on the host as well as the occurrence of the infecting agent. As a result, metabolomic studies may offer a set of markers that are useful for rapid tests to confirm infection by COVID-19, the disease severity, and the likelihood of a positive outcome. Metabolomics has been ongoing in various studies, especially examining COVID-19 infection in humans [126,127]. Understanding COVID-19’s impact on host metabolism is still important to better understand the variety of clinical presentations and provide better treatments for those affected. Metabolic profiling can find biomarkers unlike RT-PCR and can be utilized as a diagnostic and a prognostic approach, which is essential for predicting future epidemics, particularly in COVID-19 scenarios [128]. It has been remarkable how quickly COVID-19 has carried out its genome-wide association studies (GWAS), due in part to the collaborative networks set up during previous GWAS and the use of previously genotyped study populations such as UK Biobank and AncestryDNA [129,130,131,132]. GRASP portal Covid-19 GWAS results reveal potential SARS-CoV-2 modifiers [133].

Urinary metabolites from several biochemical pathways distinguish between AKI and non-AKI in COVID-19-infected individuals hospitalized. They point to a conserved defect in NAD^+^ production as a possible new therapeutic target to treat AKI brought on by COVID-19 [134]. ADAM17, a disintegrin and metalloproteinase 17 is identified as the protease responsible for the shedding of ACE2, which is the cellular receptor of SARS-CoV [135]. The proteolytic action of ADAM17 also releases the soluble forms of TNF and its receptors TNFR1 and TNFR2, which are pro-inflammatory molecules [136]. Increased expression of TNFR1 and ACE2 will worsen the prognosis of COVID-19 [137]. The downregulation of ACE2 upon infection would enhance the renal levels of Ang-II and the oxidative stress induced by Ang-II, resulting in increased renal injury [138]. According to the study of Vergara et al., urine ACE2 (uACE2) is increased in COVID-19 patients. It increased far more in patients with AKI and strongly correlated with the TNFR1 and uTNFR2. In kidney sections, elevated uACE2 was associated with tubular loss of ACE2. Increased excretion of amino acids such as tryptophan, leucine, isoleucine, and phenylalanine was observed in the urine metabolome study of the COVID-19 patients, which suggested a strong correlation between uACE2 and urine amino acids [139].

### 4.1. Untargeted Metabolomics in COVID-19 and Other Metabolomics Technologies

Global detection and relative quantification of small molecules in a given sample is the primary focus of untargeted metabolomics, while targeted, in contrast, focuses on quantifying specific groups of metabolites and offers the possibility of absolute quantification [140,141]. Untargeted metabolomics often compares the metabolomes of the experimental samples and control groups to find differences in the metabolite profiles of the two groups. These metabolomic variations could be significant for particular biological circumstances [142,143].

An untargeted metabolomic study was conducted by Chen et al., where they investigated the difference between the metabolites present in the serum of 20 healthy and 20 COVID-19-affected patients with the help of high-resolution UHPLC-MS/MS [144]. 714 metabolites were identified in the study, and around 203 metabolites were found to be different compared to healthy and infected samples. Another untargeted metabolomic analysis was performed using the saliva samples collected from the infected patients. There were statistically significant changes in a number of the detected metabolites between people with high and low severity [145]. Another prior research discovered that salivary 2-pyrrolidineacetic acid and Myo-inositol could discriminate between inpatient and outpatient cohorts [146]. A metabolomic analysis has been used to screen for metabolic changes and to give a thorough understanding of endogenous metabolites. The untargeted metabolomics study identified 2466 metabolite peaks totaling 631 and 1835 differential metabolites in negative and positive ion modes, respectively. Among 240 metabolites, 193 were substantially linked to COVID-19 [147].

As a result of an integrated analysis of 139 COVID-19 patients’ clinical measurements, immune cells, and plasma multi-omics, Su et al. identified a major change between mild and moderate COVID-19 disease, at which point inflammatory signaling is elevated and specific metabolites and metabolic processes are lost. A single axis of immune features was condensed from 120,000 immune features and aligned separately with changes in plasma composition, clinical measures of blood clotting, and transitions between mild and moderate disease to illustrate how different immune cell classes coordinate in response to SARS-CoV-2 [148]. Single-cell metabolomics is an emerging technology in metabolomics including mass-spectrometry-based single-cell metabolomics [149,150], microfluidic-based single-cell metabolomics [151], supermolecular probe based metabolic uptake competition assay [152], or Raman spectroscopy based single-cell or even subcellular metabolomics [153,154]. These are four different categories of methods that could be complementary to each other. A multi-omic analysis of single cells can also be done by integrating single-cell metabolic analysis with other omics, such as proteomics [148,155]. These techniques could be useful due to its potential applications in COVID-19 and kidney disease.

### 4.2. Antiviral Drug Efficacy in COVID-19

Several experimental methods, such as repurposing RNA polymerase-inhibiting antiviral medications, have improved the health outcomes for COVID-19 patients. To combat COVID-19, effective antiviral medicines are crucial. There has been evidence that lopinavir and various IFs, notably IF- β, have moderate anti-SARS-CoV efficacy in vitro [156]. It was also noted that another drug called ribavirin, showed synergistic effects against COVID-19 [157]. It was also demonstrated that lopinavir-ritonavir or IF-β1b could lower viral load and improve lung histology [158]. Intranasal leukocytic IF-α or IF-β1a are expected to be useful for prophylaxis against SARS without a successful vaccination. The best combination appears to be IF-α, IF- β1a, and ribavirin. A combination with a brief course of ribavirin seems appropriate since IFs might not be efficient in causing an antiviral response in the uninfected host cells within the first 24 h [157].

A computational study analyzed commonly used drugs, including Favipiravir, Remdesivir, Nitazoxanide, Galidesivirm, and Ribavirin [159]. Monitoring the bioenergetic state of patients may aid in explaining why some patients respond well to a specific replicase-transcriptase inhibitor while others do not. According to this viewpoint, medications with high ATP reliance will be less successful in treating patients with advanced metabolic dysfunction. Therefore, it is suggested that individuals with a near-normal metabolic profile might have a greater probability of responding to ribavirin or favipiravir. These medicines need multiple-stage functionalization [160]. Many drugs used to treat COVID-19 are known to cause AKI [161] (Table 1).

### 4.3. Metabolites Involved in the Diagnosis and Prognosis of COVID-19

Several pilot experiments were carried out to determine and identify the COVID-19 connected to volatile organic compound (VOC) indicators and assess the potential for COVID-19 testing in contrast to conventional RT-qPCR. In an untargeted metabolomic study, VOCs including methylpent-2-enal, 1-chloroheptane, 2,4-octadiene and nonanal in the breath of COVID-19 patients were found to be elevated significantly [26]. The initiation of the immunological responses in COVID-19 includes several metabolic pathways, including the metabolism of amino acids, energy, and lipids. Arachidonic acid is an inherently bioactive antiviral lipid, and it has been hypothesized that this metabolic route significantly impacts COVID-19 susceptibility [169]. It was also observed that in the pathogenesis of COVID-19, cyclooxygenase-2 (COX-2) and prostaglandins, notably PGE2, have pro-inflammatory effects. Additionally, hybrid drugs, such as COX-2 inhibitors, can potentially treat patients with COVID-19 by regulating the total balance of arachidonic acid mediators [170].

Profiling small metabolites and macromolecules allow measuring the host’s response to the infection. A human antiviral metabolite, 3′-Deoxy-3′,4′-didehydro-cytidine, was shown to be considerably higher in COVID-19 patients [171]. Compared to the non-COVID-19 and healthy groups, the 15-HETE levels identified in the COVID-19 individuals were significantly depleted. The lack of anti-inflammatory signals brought on by the reduction of 15-HETE may be a factor in the increased inflammation observed during COVID-19 infection [172]. A targeted metabolomic study analyzed and identified AMP, dGMP, sn-glycerol-3-phosphocholine, and carnitine metabolites that were dysregulated in patients with COVID-19 [173]. The study by Lee et al. analyzing plasma metabolite and protein levels, as well as single-cell multi-omics analyses collected during the first week after clinical diagnosis, metabolic changes associated with peripheral immune responses in 198 COVID-19 patients were reported in a large cohort study of healthy donors. Plasma metabolites like acetoacetate, which is formed in response to poor cellular glucose uptake, and α-ketobutyrate, which is implicated in the development of an early insulin resistance biomarker, α-hydroxybutyrate, can be used to predict future outcomes of newly diagnosed COVID-19 patients. These metabolites are well-known in other disorders with COVID-19-like pathogenesis. It may be possible to predict the survival rates by combining the measurement of these plasma metabolites with cell-type-specific metabolic reprogramming networks that are linked to disease severity [174]. More studies are required to analyze and understand different metabolites to help diagnose the infected patients faster.

### 4.4. Alteration of Plasma Metabolomics in COVID-19

According to the level of immunological response, different plasma metabolic profiles were seen, indicating the importance of the metabolism of amino acids and the lipid profile acting as indicators of how well a vaccine will work. Alterations in metabolomics in the plasma were detected in different studies. In one such study, alterations in plasma were analyzed, and changes in different amino acids and lipid profiles were detected as a response to vaccination against COVID-19 [175]. COVID-19 patients’ lipid and metabolite changes correlate with their disease progression, suggesting that COVID-19 impacted their entire body metabolism. Particularly, altered energy metabolism and hepatic dysfunction are brought on by malic acid of the TCA cycle as well as the carbamoyl phosphate of the urea cycle [176]. Another metabolomic pathway analysis revealed that the individuals affected with COVID-19 were found to have an impact on glycerophospholipid and the porphyrin metabolism but at the same time showed a significant effect on the glycerophospholipid and linoleic acid metabolism pathways [177]. Shen et al., analyzed the proteome and metabolome of the sera from several COVID-19 patients by applying proteomics and metabolomics to test whether any characteristic molecular changes are induced by SARS-CoV-2. Proteomic and metabolomic studies revealed molecular changes in the sera of 46 COVID-19 patient group compared to the 53 individuals in the control group suggesting dysregulation of macrophages, degranulation of platelets, massive metabolic suppression and complement system pathways in the COVID-19 infected group. By molecular signatures of metabolites and proteins using a machine learning model based on the expression levels of 22 serum proteins and 7 metabolites, COVID-19 severity cases can be classified [178].

## 5. Integrated Genomics and Metabolomics in Nephrology

In contrast to estimated glomerular filtration rate (eGFR), a traditional marker, kidney biopsies, urine, and blood samples can be utilized to develop metabolomics, genomics, transcriptomics, and proteomic biomarkers. These indicators may be more strongly and precisely linked to the pathophysiologic mechanisms of the disease [179,180,181]. Genes and metabolites related to renal pathophysiology have been found using ‘omics’ techniques, such as metabolome analysis and GWAS. From meta-analyses of GWAS from large epidemiologic populations, several novel loci connected with the eGFR and CKD have been discovered. [182]. The implications associated with different stages of CKD include changes in the steroid hormone, glucose, NO, purine, and lipid metabolism [183,184]. The clinical utility of metabolomics in pediatric nephrology includes the identification of biomarkers, which are as of yet unidentified biological therapeutic targets, the linking of metabolites to pertinent standard indices and clinical outcomes, and the opportunity to study the intricate interactions between genetic and environmental factors in particular disease states [185]. The mechanism of metabolites and metabolic ratios associated with alteration in kidney functions are an accumulation of metabolites in response to impaired renal function mainly in the tubular cells, urine, or blood, e.g., creatinine, metabolites reflecting the activity of enzymes which are expressed in the kidney tissues and metabolites which contribute directly to disease progression [186].

### 5.1. CKD

The identification of monogenic causes of CKD has been accomplished through genomic profiling, with approximately 500 genes identified so far, the majority of which being reported in pediatric patients [187]. Approximately up to 37% of the adult cases among the adult population are attributed to inheritable kidney disease [188,189]. Numerous GWAS studies have been carried out, expanding our understanding of the genetic variations associated with CKD and the surrogate measure for renal function, eGFR [190]. One such GWAS study by Köttgen et al., conducted in 19,877 European individuals to identify the susceptibility loci for glomerular filtration by estimating the serum creatinine and cystatin C and CKD identified significant associations of single nucleotide polymorphism with CKD at the UMOD locus [191]. In a follow-up study, he confirmed that the elevated levels of uromodulin are linked with a common polymorphism in the UMOD region and are linked with the onset of CKD, and rs4293393 can be one of the factors related to the altered concentrations of uromodulin [192]. Indoleamine 2,3-dioxygenase (IDO), a crucial factor in T-cell inactivation, is increased by tryptophan and an increase in kynurenic acid and kynurenine. A tryptophan metabolite produced by gut bacteria called indoxyl sulphate downregulates renal-specific organic anion transporter (OAT) SLCO4C1. It has been linked to tubular cell failure [193].

### 5.2. Diabetic Nephropathy

One of the complications arising from type 1 diabetes (T1D) and T2D is diabetic nephropathy (DN). Its known metabolomic biomarkers include ketone bodies (3-hydroxybutyrate), sugar metabolites (1,5-anhydroglucoitol), free fatty acids, and branched-chain amino acids [194]. Engulfment and cell motility 1 gene (ELMO1) is a potential candidate for causing DN, according to the results of the GWAS in Japanese patients [195]. Salem et al., in their study, identified 16 genome-wide significant risk loci in a population of about 20,000 European descent with type 1 diabetes [196]. The characteristics of DN, such as expansion and thickening of the glomerular basement membrane, resulting from increased expression of extracellular matrix protein and ELMO1 activity [197].

## 6. Therapeutic Management of Renal Disorder Patients with COVID-19

### 6.1. Vitamins

Vitamin D deficiency is seen in most patients suffering from CKD; its deficiency can affect the COVID-19 outcome. Vitamin D influences innate and adaptive immunity, which influences how the immune system reacts to viruses and bacteria [198]. According to previous research, even a short-term acute vitamin D shortage can cause hypertension and affect the parts of the renin-angiotensin system that cause kidney damage. Vitamin D insufficiency has repeatedly been linked to proteinuria, albuminuria, end-stage renal disease (ESRD) development, and a higher risk of death from all causes in those with CKD [199]. In some instances, anemia developed in dialysis patients can be managed by giving supplemental vitamin C intravenously or orally [200]. Reactive oxygen species (ROS) are quenched by the immune cells, which otherwise may damage the lungs. Vitamin C deficiency has been linked to weakened immunity and increased susceptibility to diseases. In symptomatic COVID-19 patients, having a strong antioxidant system to eliminate such excess ROS might be beneficial [201]. 500 mg/kg of vitamin E supplementation is found to inhibit ferroptosis in COVID-19 patients and reduce the damage that ferroptosis causes to various organs, such as the heart, liver, kidneys, stomach, and neurological system leading to inflammation ablation and viral clearance via the modulation of T cells [202].

### 6.2. Metal Supplements

In the early stages of COVID-19 infection, adding metal nutrients can enhance good immune function and be a prophylactic measure for high-risk individuals. Metal nutrients may help lower COVID-19 infection rates and rates of severe disease and fatality [203]. Zinc (Zn) has been identified to prevent the synthesis of RNA, which prevents the entry, fusion, replication, protein translation, and viral propagation of many RNA viruses [204,205,206,207]. Apart from this, zinc, when administered in the form of zinc gluconate, has been shown to reduce the duration of common cold when administered within 24 h of the onset of symptoms [207]. People with CKD, especially with nephrotic disease and uremia, have been shown to have abnormalities in Zn metabolism; this alteration in the metabolism of Zn may be due to less intake of Zn in the diet, poor intestinal absorption, increased endogenous secretion and increased urinary excretion of Zn. The reason for the deficiency of Zn in people with kidney diseases is unclear [208]. Selenium (Se) deficiency is seen in AKI or CKD and COVID-19; its deficiency also has detrimental effects in the case of various viral disorders. Supplementation of Se reduces the progression of COVID-19. In individuals with CKD, it increases the GSH-Px, which plays an essential role in the metabolism of ROS [209,210]. In obese COVID-19 patients, reduced renal function and lower magnesium levels were linked to increased mortality rates [211]. Magnesium supplements are administered to patients with AKI and COVID-19 patients with other comorbidities [212,213,214].

### 6.3. Melatonin

Melatonin is a well-known antioxidative and anti-inflammatory molecule. It is used in the treatment of critical care patients to reduce anxiety, vessel permeability, use of sedation, and improve sleep quality [30]. The cost-effectiveness, minor side effects, and properties of melatonin, such as anti-viral properties, an antioxidant enzyme inducer, apoptosis regulator, immune function stimulator, and free radical scavenger, make it a potential adjuvant in the treatment of COVID-19 patients and other viral infections [215,216,217]. Melatonin prevents fibrosis which is a complication of COVID-19. The inflammasome activity required for COVID-19 to cause lung inflammation is blocked by melatonin [218]. Through receptor-mediated or receptor-independent biological actions, melatonin regulates mitochondrial metabolism, boosts ATP production, guards’ mitochondria from nitrative damage, and has pluripotent protective effects in the kidneys. By inactivating free radicals by the transfer of one or more electrons, it effectively reduces oxidative stress. Additionally, it reduces pro-inflammatory mediators and initiates a preventive mechanism against inflammation-related chronic injury [167,219]. Melatonin reduces kidney damage caused due to COVID-19 drugs such as the lopinavir/ritonavir combination [220].

### 6.4. Renal Replacement Therapy

About 64% of critically ill COVID-19 individuals with AKI require renal replacement therapy (RRT) as it results from aberrant electrolyte concentrations and resistance to pharmacological therapy for volume overload [34,221,222,223]. RRT initiation in AKI patients with complications of COVID-19 alleviates not only AKI but also COVID-19. Without critical conditions, they have no significant effect on mortality and renal recovery [224]. As a result, in patients with COVID-19-associated AKI, RRT has been widely recommended when metabolic. Fluid demands exceed total kidney capacity instead of strictly determining the need for RRT initiation based on blood urea nitrogen or creatinine levels [42]. Continuous RRT is suggested in COVID-19 patients with hemodynamic instability [223]. However, its use requires extensive training, higher complexity, time consumption, continuous anticoagulation, multiple equipment, and higher cost [225,226]. COVID-19 patients with AKI are also suggested prolonged intermittent renal replacement therapy (PIRRT) due to its viability, safety, lower cost, less nursing time, flexible treatment schedules, and acceptable hemodynamical tolerability [227].

## 7. Conclusions

Various pathological findings can accompany SARS-CoV-2 infection, and an AKI condition directly caused by a virus is possible but not widespread. We have a unique opportunity to develop novel therapies to delay the progression of kidney disease by understanding how COVID-19-associated AKI might worsen CKD. Patients with COVID-19 AKI have frequently shown signs of endothelial damage, microvascular thrombi, local inflammation, and immune cell infiltration; nevertheless, it is unclear whether the etiologies of COVID-19 AKI and non-COVID-19 sepsis-associated AKI differ or are comparable. It is also unclear whether direct viral infection contributes to AKI development. The exact mechanism by which COVID-19 causes kidney damage remains unclear.

## Figures and Tables

**Figure 1 vaccines-11-00489-f001:**
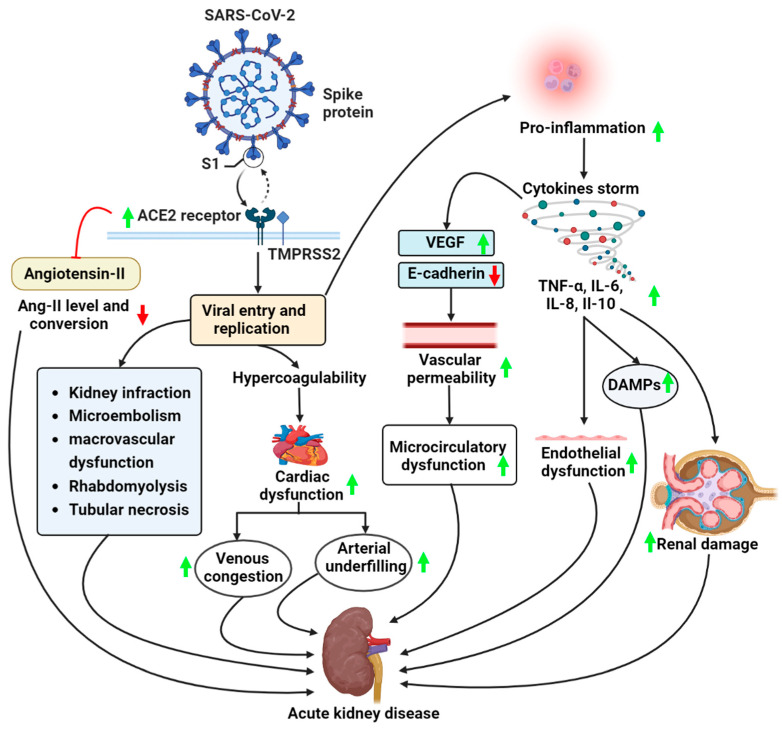
This illustration shows the pathogenesis of COVID-19-associated acute kidney disease (AKD). The receptor binding domain (RBD) of the virus binds to the ACE2 receptor. TMPRSS2 proteases cleave the spike protein of the virus. This supports viral entry from the plasma membrane and replication of the virus, resulting in increased pro-inflammation and cytokine storms, leading to microcirculatory dysfunction, endothelial dysfunction, and renal damage. However, viral entry and replication also lead to increased hypercoagulation, which ultimately leads to increasing cardiac dysfunction. This results in increasing venous congestion and arterial underfilling, which leads to AKD. (Red arrow-Downregulation; Green arrow-Upregulation). (Abbreviations: VEGF, vascular endothelial growth factor; TNF-α, tumor necrosis factor-alpha; IL-6, interleukin-6; IL-8, interleukin-8; IL-10, interleukin-10; DAMPs, damage-associated molecular patterns).

**Table 1 vaccines-11-00489-t001:** COVID-19 drugs causing kidney injury.

Drug Class	Drug	Study Model	Kidney Injury	Mechanism	Reference
Antiviral	Remdesivir	Humans	AKI	↑ proximal tubular epithelial cell necrosis	[162]
	lopinavir/ritonavir	Humans	AKI	↓ glomerular filtration rate, proteinuria, glycosuria	[163]
	Favipiravir	Humans	AKI	↑ hyperuricemia	[164]
	Tenofovir	Humans	Subclinical tubular injury	↑ proteinuria	[165]
Antimalarial	Hydroxychloroquine and chloroquine	In-vivo: female C57BL/6JOlaHsd mice In-vitro: U2OS, HeLa cells, HeLa-RFP-GFP-LC3 cells, SNAP29- and GFP-STX17-expressing MEFs, MEFs	AKI	↓ autophagy, ↑ lysosomal pH, mitochondrial damage	[166]
Antibiotic	Azithromycin	Humans	AKI	Induce interstitial nephritis	[167]
	Amphotericin B	Humans	AKI	Arteriolar vasoconstriction; direct tubular injury	[168]

## Data Availability

Data are available from the authors on request (A.V.G.).

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
