# Peer review of "Crosstalk between COVID-19 Infection and Kidney Diseases: A Review on the Metabolomic Approaches"

_vaccines, 2023, doi:10.3390/vaccines11020489_

Round 1

Reviewer 1 Report

Reshma Murali et al, provided a systematic review of the crosstalk between COVID-19 and Kidney disease within the context of metabolomics approaches. Overall, this paper is thorough in covering many of the important aspects of metabolomics and their covid-related or kidney disease-related applications. However, a few important aspects were missed and some key references will need to be included. Thus, a revision addressing all of the following issues would be needed.

1.       In discussing the “Alteration of plasma metabolomic in COVID-19”, the authors missed a few key large-scale studies, for example, PMID: 32492406, which even integrated proteomics with metabolomics. The authors should be more inclusive in their literature search in order to not miss at least those large-scale studies.

2.       In discussing the “Metabolites involved in the diagnosis and prognosis of COVID-19”, the authors missed a few key large-scale studies, for example, PMID: 34489601, which even found specific metabolites that can predict survival and validated in a second cohort. The authors should be more thorough in their literature search in order to avoid missing at least these large-scale studies.

3.       Connections of metabolomics approach to “Long covid” especially kidney-related long-covid symptoms, is another important topic that is, at least, worthwhile mentioning, even if it may not comprise a majority of the content, given the potentially huge number of populations that are influenced by “Long covid”. For example, PMID: 34425843, 35216672, etc.

4.       Metabolomic and its integration with other omics to study COVID-19 or kidney disease is worth a separate section. The authors have already discussed the integration with genomics, which is a very good angle. However, its integration with immune omics is still lacking. For example, in PMID: 33171100, the authors integrated plasma metabolomics with proteomics and immune cell transcriptomics at single-cell resolution to reveal a coherent multi-omic module that explains the sharp disease transition between mild and moderate covid-19 across multiple layers of omics, indicating the crosstalk between metabolomics and other immune response and functions.

5.       The authors covered some metabolomics technologies including untargeted metabolomics. However, one of the key emerging category of technologies for the field of metabolomics, is single-cell metabolomics. Mass-spectrometry-based single-cell metabolomics (for example PMID: 31565762, 33595331), microfluidic-based single-cell metabolomics (for example PMID: 32393797 ), supermolecular probe based metabolic uptake competition assay (for example Pubmed: 26916347), or Raman spectroscopy based single-cell or even subcellular metabolomics (for example 33595331) are three different categories of methods that could be complementary to each other. Furthermore, such single-cell metabolic analysis has even been integrated with other omics such as proteomics to perform multi-omic measurements from the same single cells (for example PMID: 33171100, 29992724). The paper will miss such a great opportunity if not cover this emerging field of metabolomics and their potential applications in covid-19 and kidney disease contexts.

Author Response

Reviewer 1

Comments and Suggestions for Authors

Reshma Murali et al, provided a systematic review of the crosstalk between COVID-19 and Kidney disease within the context of metabolomics approaches. Overall, this paper is thorough in covering many of the important aspects of metabolomics and their covid-related or kidney disease-related applications. However, a few important aspects were missed and some key references will need to be included. Thus, a revision addressing all of the following issues would be needed.

Response: All the authors are thankful to the reviewer for evaluating the quality of the manuscript and for potential comments to maintain the standard of the manuscript. We have addressed all the comments in the main track changed manuscript.

  1. In discussing the “Alteration of plasma metabolomic in COVID-19”, the authors missed a few key large-scale studies, for example, PMID: 32492406, which even integrated proteomics with metabolomics. The authors should be more inclusive in their literature search in order to not miss at least those large-scale studies.

Response: We have included the suggested study in section 4.4. Alteration of plasma metabolomic in COVID-19. The track changes are made in the main manuscript (Line number 473-482)

Shen et al., analyzed the proteome and metabolome of the sera from several COVID-19 patients by applying proteomics and metabolomics to test whether any characteristic molecular changes are induced by SARS-CoV-2. Proteomic and metabolomic studies revealed molecular changes in the sera of 46 COVID-19 patient group compared to the 53 individuals in the control group suggesting dysregulation of macrophages, degranulation of platelets, massive metabolic suppression and complement system pathways in the COVID-19 infected group. By molecular signatures of metabolites and proteins using a machine learning model based on the expression levels of 22 serum proteins and 7 metabolites, COVID-19 severity cases can be classified [1].  

  1. In discussing the “Metabolites involved in the diagnosis and prognosis of COVID-19”, the authors missed a few key large-scale studies, for example, PMID: 34489601, which even found specific metabolites that can predict survival and validated in a second cohort. The authors should be more thorough in their literature search in order to avoid missing at least these large-scale studies.

Response: We have included the suggested study in section 4.3. Metabolites involved in the diagnosis and prognosis of COVID-19. The track changes are made in the main manuscript (Line number 448-458)

The study by Lee et al. analyzing plasma metabolite and protein levels, as well as single-cell multiomics analyses collected during the first week after clinical diagnosis, metabolic changes associated with peripheral immune responses in 198 COVID-19 patients were reported in a large cohort study of healthy donors. Plasma metabolites like acetoacetate, which is formed in response to poor cellular glucose uptake, and α-ketobutyrate, which is implicated in the development of an early insulin resistance biomarker, α-hydroxybutyrate, can be used to predict future outcomes of newly diagnosed COVID-19 patients. These metabolites are well-known in other disorders with COVID-19-like pathogenesis. It may be possible to predict the survival rates by combining the measurement of these plasma metabolites with cell-type-specific metabolic reprogramming networks that are linked to disease severity [2].

  1. Connections of metabolomics approach to “Long covid” especially kidney-related long-covid symptoms, is another important topic that is, at least, worthwhile mentioning, even if it may not comprise a majority of the content, given the potentially huge number of populations that are influenced by “Long covid”. For example, PMID: 34425843, 35216672, etc.

Response: We have included the suggested studies about long-covid in the introduction section. The track changes are made in the main manuscript (Line number 71-87)

Pretorius et al. reported that in COVID-19 patients who have recovered from acute COVID-19 symptoms, a novel phenotype known as Long COVID/Post-Acute Sequelae of COVID-19 (PASC) has been identified. Proteomics research revealed large increases in acute phase proteins such as SAA4 and α2AP, indicating that the plasma samples from COVID-19 and PASC patients are highly resistant to breakdown in the presence of trypsin [3]. According to Al-Aly et al. study that evaluated lengthy COVID using electronic health data from the Veterans Health Administration (VHA), COVID-19 increased the risk of CKD and that risk was highest in patients with severe disease [4]. Su et al. carried out a comprehensive multi-omic longitudinal study of 309 COVID-19 patients from the time of their first diagnosis to convalescence (2–3 months later), integrating clinical information and patient-reported symptoms and four PASC-anticipating risk factors were resolved upon diagnosis of COVID-19: type 2 diabetes, SARS-CoV-2 RNAemia, Epstein-Barr virus viremia, as well as specific autoantibodies. SARS-CoV-2 and CMV-specific CD8+ T lymphocytes in individuals with gastrointestinal PASC displayed unique dynamics during COVID-19 recovery. Symptom-associated immunological signatures reveal four endotypes with divergent acute severity and PASC [5].

  1. Metabolomic and its integration with other omics to study COVID-19 or kidney disease is worth a separate section. The authors have already discussed the integration with genomics, which is a very good angle. However, its integration with immune omics is still lacking. For example, in PMID: 33171100, the authors integrated plasma metabolomics with proteomics and immune cell transcriptomics at single-cell resolution to reveal a coherent multi-omic module that explains the sharp disease transition between mild and moderate covid-19 across multiple layers of omics, indicating the crosstalk between metabolomics and other immune response and functions.

Response: We have included the suggested study in section 5. Integrated genomics and metabolomics in nephrology. The track changes are made in the main manuscript (Line number 389-396).

As a result of an integrated analysis of 139 COVID-19 patients' clinical measurements, immune cells, and plasma multi-omics, Su et al. identified a major change between mild and moderate COVID-19 disease, at which point inflammatory signaling is elevated and specific metabolites and metabolic processes are lost. A single axis of immune features was condensed from 120,000 immune features and aligned separately with changes in plasma composition, clinical measures of blood clotting, and transitions between mild and moderate disease to illustrate how different immune cell classes coordinate in response to SARS-CoV-2 [6].   

  1. The authors covered some metabolomics technologies including untargeted metabolomics. However, one of the key emerging category of technologies for the field of metabolomics, is single-cell metabolomics. Mass-spectrometry-based single-cell metabolomics (for example PMID: 31565762, 33595331), microfluidic-based single-cell metabolomics (for example PMID: 32393797 ), supermolecular probe based metabolic uptake competition assay (for example Pubmed: 26916347), or Raman spectroscopy based single-cell or even subcellular metabolomics (for example 33595331) are three different categories of methods that could be complementary to each other. Furthermore, such single-cell metabolic analysis has even been integrated with other omics such as proteomics to perform multi-omic measurements from the same single cells (for example PMID: 33171100, 29992724). The paper will miss such a great opportunity if not cover this emerging field of metabolomics and their potential applications in covid-19 and kidney disease contexts.

Response: We have included all the suggested techniques in a single paragraph in section 4.1. Untargeted metabolomics in COVID-19 and other metabolomics technologies. The track changes are made in the main manuscript. (Line number 394-401).

Single-cell metabolomics is the emerging technologies in metabolomics including mass-spectrometry-based single-cell metabolomics [7, 8], microfluidic-based single-cell metabolomics [9], supermolecular probe based metabolic uptake competition assay [10], or Raman spectroscopy based single-cell or even subcellular metabolomics [8]. These are three different categories of methods that could be complementary to each other. A multi-omic analysis of single cells can also be done by integrating single-cell metabolic analysis with other omics, such as proteomics [6, 11]. These techniques could be potential applications in covid-19 and kidney disease contexts.

Reviewer 2 Report

This review article collected references to describe the crosstalk between COVID-19 infection and kidney diseases. The authors tried to focus on metabolomic studies in COVID-19-mediated kidney diseases. However, most results were from COVID-19 patients and independent of kidney diseases.

Major comments:

(1)   It is difficult to read due to poor organization. I suggested reorganizing the description of AKI, CKD, and metabolomics.

(2)   The last paragraph of the Introduction section should clearly mention the specific aims of this review, and what is different with other reviews discussing COVID-19 and kidney diseases.

(3)   There are some terms in Figure 1 not described in the text, including macrovascular dysfunction, hypercoagulability, cardiac dysfunction, venous congestion, arterial underfilling, VEGF, E-cadherin, vascular permeability, and microcirculatory dysfunction.

(4)   Section 4.2 and Table1, why did not include the common anti-COVID-19 drugs Paxlovid and Molnupiravir? Do they cause AKI?

Minor comments:

(1) lines 84 and 86, reverse ICU and intensive care units.

(2) Lines 91-92, please mention the situations of 0.5% to 80%, which is a huge difference. Lines 190-191, why is 4-7%?

(3) Line 96, Acute kidney injury should change to AKI.

(4) Line 121, please define sACE2.

(5) Lines 154-155, “As ACE2 is a negative regulator of ACE”?

(6) Lines 165-173, redundant information and unrelated to Rhabdomyolysis!

(7) Lines 181-183, please cite the reference.

(8) Line 245, please define APOL1.

Author Response

Reviewer 2

Comments and Suggestions for Authors

This review article collected references to describe the crosstalk between COVID-19 infection and kidney diseases. The authors tried to focus on metabolomic studies in COVID-19-mediated kidney diseases. However, most results were from COVID-19 patients and independent of kidney diseases.

Response: All the authors are thankful to the reviewer for evaluating the quality of the manuscript and for potential comments to maintain the standard of the manuscript. We have addressed all the comments in the main track changed manuscript.

Major comments:

(1)   It is difficult to read due to poor organization. I suggested reorganizing the description of AKI, CKD, and metabolomics.

Response: We have reorganized the sequence of AKI, CKD and metabolomics and track changes are made in the manuscript (Line number: 51-94).

(2)   The last paragraph of the Introduction section should clearly mention the specific aims of this review, and what is different with other reviews discussing COVID-19 and kidney diseases.

Response: Authors are thankful for your suggestion. Accordingly, we have made changes in the main manuscript (Line number: 95-98).

The aim of this narrative review is to discuss the possible applications of analytical strategies such as metabolomics in the study of COVID-19, kidney and COVID-19 induced kidney injury and to address the therapeutic options to treat the COVID-19 induced kidney injury.

(3)   There are some terms in Figure 1 not described in the text, including macrovascular dysfunction, hypercoagulability, cardiac dysfunction, venous congestion, arterial underfilling, VEGF, E-cadherin, vascular permeability, and microcirculatory dysfunction.

Response: Authors are thankful to the reviewer for the comment. This figure provides a concise, clear, easy-to-understand means of helping the reader to grasp important concepts and pathophysiology. We have given the detail explanation of figure in the figure legend covering all the terminologies. Whereas, most of the terminology explained in section 2.2. Pathophysiology of COVID-19 and AKI.

(4)   Section 4.2 and Table1, why did not include the common anti-COVID-19 drugs Paxlovid and Molnupiravir? Do they cause AKI?

Response: Studies available suggest that Paxlovid is efficient in high-risk patients with AKI, decreasing the length of hospital stay for AKI patients who developed COVID-19. Molnupiravir is also used in end stage kidney disease patients undergoing hemodialysis or kidney transplantation during the onset of SARS-COV-2. Hence proper evidence on whether they cause kidney injury couldn’t be found.

Minor comments:

(1) lines 84 and 86, reverse ICU and intensive care units.

Response: We have made the changes and maintain the consistency throughout the manuscript.

(2) Lines 91-92, please mention the situations of 0.5% to 80%, which is a huge difference.

Response: We have removed this line to prevent any confusion.

Lines 190-191, why is 4-7%?

Response: We have removed this line to prevent any confusion.

(3) Line 96, Acute kidney injury should change to AKI.

Response: We have made the changes and maintain the consistency throughout the manuscript.

(4) Line 121, please define sACE2.

Response: We have corrected the abbreviation as recombinant human soluble ACE2.

(5) Lines 154-155, “As ACE2 is a negative regulator of ACE”?

Response:  Yes, ACE2 has a multiplicity of physiological roles that revolve around its function and acts as a negative regulator of ACE.

(6) Lines 165-173, redundant information and unrelated to Rhabdomyolysis!

Response: We have removed the unrelated data from the section 2.1.3. Rhabdomyolysis as per the suggestion.

(7) Lines 181-183, please cite the reference.

Response:  We have given citation to the suggested line is as (Line number: 193-195)

Recent research indicates that the viral invasion of myocytes during influenza infection is the direct cause of rhabdomyolysis [80].

(8) Line 245, please define APOL1.

Response: We have given the full form for APOL1 is Apolipoprotein L1 (APOL1) in the main manuscript.

Reviewer 3 Report

Review of vaccines-2192283

Crosstalk between COVID-19 Infection and Kidney Diseases: A review on the Metabolomic Approaches

Reshma Murali, Uddesh Ramesh Wanjari, Anirban Goutam Mukherjee, Abilash Valsala Gopalakrishnan, Sandra Kannampuzha, Arunraj Namachivayam, Harishkumar Madhyastha, Kaviyarasi Renu and Raja Ganesan

General comments

This manuscript describes the association between COVID-19 and kidney disease. This paper is an important contribution. Contrary to the title, the main context is not focused on metabolism. It is difficult for readers to follow the text because it goes off the topics. In my opinion, a substantial revision is needed to make this manuscript suitable for publication.

Specific comments

1. Why did the authors include “Rhabdomyolysis”? There are many reasons for triggering AKI. It is somewhat odd that the authors point only to rhabdomyolysis.

2. Do the authors think the section “5. Integrated genomics and metabolomics in nephrology” really is necessary? If the authors include genomics, the authors would include the results of GWAS of COVID-19.

3. Why did the authors divide metabolomics into two sections?

“4. Metabolomic and COVID-19” and “6. Metabolomics and COVID-19-mediated Kidney”

4. In “2. COVID-19 and AKI” section, there are mix of US and USA. The authors would unify this.

5. In line 119-120, is the “elevated angiotensin II” correct?

Author Response

Reviewer 3

Comments and Suggestions for Authors

Review of vaccines-2192283

Crosstalk between COVID-19 Infection and Kidney Diseases: A review on the Metabolomic Approaches

Reshma Murali, Uddesh Ramesh Wanjari, Anirban Goutam Mukherjee, Abilash Valsala Gopalakrishnan, Sandra Kannampuzha, Arunraj Namachivayam, Harishkumar Madhyastha, Kaviyarasi Renu and Raja Ganesan

General comments

This manuscript describes the association between COVID-19 and kidney disease. This paper is an important contribution. Contrary to the title, the main context is not focused on metabolism. It is difficult for readers to follow the text because it goes off the topics. In my opinion, a substantial revision is needed to make this manuscript suitable for publication.

Response: All the authors are thankful to the reviewer for evaluating the quality of the manuscript and for potential comments to maintain the standard of the manuscript. We have addressed all the comments in the main track changed manuscript.

Specific comments

  1. Why did the authors include “Rhabdomyolysis”? There are many reasons for triggering AKI. It is somewhat odd that the authors point only to rhabdomyolysis.

Response: There was rhabdomyolysis in a severe COVID-19 patient accompanied by fatigue and pain in the lower limbs. Damaged cells release myoglobin, causing AKI, a serious complication of severe rhabdomyolysis. AKI caused by COVID-19 infection results in hemosiderin deposition and pigment casts in tubules, which confirms rhabdomyolysis.

For this reason, we have included the Rhabdomyolysis. Also, we have included section 2.1.4. Sepsis which also shares similarities with COVID-19 AKI, which is of interest. (Line numbers: 213-223)

  1. Do the authors think the section “5. Integrated genomics and metabolomics in nephrology” really is necessary? If the authors include genomics, the authors would include the results of GWAS of COVID-19.

Response: Yes, it is important topic. This topic is giving idea about integrated genomics and metabolomics studies in nephrology including CKD and Diabetic nephropathy. Summarizing the recent findings in clinical kidney diseases research revealed by ‘Omics’ approaches with a clear focus on recent genomics and metabolomics efforts.

We have included the results of GWAS of COVID-19 in the section 4. Metabolomic, COVID-19 and Kidney injury (Line numbers: 344-348).

  1. Why did the authors divide metabolomics into two sections? “4. Metabolomic and COVID-19” and “6. Metabolomics and COVID-19-mediated Kidney”

Response: We have merged these topics to one as 4. Metabolomic, COVID-19 and Kidney injury to maintain the consistency.  The track changes are made in the main manuscript.

  1. In “2. COVID-19 and AKI” section, there are mix of US and USA. The authors would unify this.

Response: We have made the necessary changes in the main manuscript and changed US to USA to maintain the consistency. The track changes are made in the main manuscript.

  1. In line 119-120, is the “elevated angiotensin II” correct?

Response: Yes, it is correct. Since ACE2 is downregulated as a result of coronavirus binding to it, angiotensin is less likely to be converted to the vasodilator angiotensin 1-7, which results in an increase in angiotensin II.

Round 2

Reviewer 1 Report

This revised manuscript is significantly improved after accounting for the suggestions from this reviewer. However, there are some references that seem to be miss-cited and thus needed to be corrected. Thus, this reviewer will suggest another (minor) revision to correct that error.

In lines 396-397, the authors correctly mentioned “the Raman spectroscopy based single cell or even subcellular metabolomics” with their reference 157. However, reference 157 is not related to Raman. A correct reference should be PMID: 32907318 and 32973134 or related.

Author Response

Reviewer:

This revised manuscript is significantly improved after accounting for the suggestions from this reviewer. However, there are some references that seem to be miss-cited and thus needed to be corrected. Thus, this reviewer will suggest another (minor) revision to correct that error.

Response: All the authors are thankful to the reviewer for this suggestion. We have addressed the comment.

In lines 396-397, the authors correctly mentioned “the Raman spectroscopy based single cell or even subcellular metabolomics” with their reference 157. However, reference 157 is not related to Raman. A correct reference should be PMID: 32907318 and 32973134 or related.

Response: All the authors are thankful to the reviewer for this suggestion. We have cited the proper references for the same in the main manuscript. (Line number 396-397)

Raman spectroscopy based single-cell or even subcellular metabolomics [160, 161].

[160]      D. Lee, J. Du, R. Yu, Y. Su, J. R. Heath, and L. Wei, "Visualizing Subcellular Enrichment of Glycogen in Live Cancer Cells by Stimulated Raman Scattering," (in eng), Anal Chem, vol. 92, no. 19, pp. 13182-13191, Oct 6 2020.

[161]      J. Du et al., "Raman-guided subcellular pharmaco-metabolomics for metastatic melanoma cells," (in eng), Nat Commun, vol. 11, no. 1, p. 4830, Sep 24 2020.

Reviewer 2 Report

The authors have improved the manuscript following my comments. I have no more suggestions. 

Author Response

All the authors are thankful to the reviewer for evaluating the quality of the manuscript. 

Reviewer 3 Report

I appreciate the authors for their collaboration.

The manuscript has been much improved and is in a nice condition now.

Author Response

(The authors gave the same response as above.)
